# Assessment of clinical medical education needs inform design of a preceptor development program in Jordan: A multi method study

Soha Albeitawi[1]*, Mohammad Talal Al-zubi[1], Anas Aljaiuossi[1], Murad Shatnawi[1], Ahlam Al-Kharabsheh[2], Fadi Sawaqed[2], Emad Aborajooh[2], Walid I. Wadi[2], Randa Mahasneh[3], Benjamin Rowland Colton[4], Mohammad AlQudah[4], Tamara Kufoof[4], Fida Asali[4], Ahmed Sheyyab[4], Monther A. Gharaibeh[4], Motasem Al-latayfeh[4], Enas Al-Zayadneh[5], Eman Badran[5], Yaser M. Rayyan[5], Kais Al Balbissi[5], Raed Al-Taher[5], Asma Basha[5], Rola Saqan[6], Ashraf Omar Oweis[6], Wafa Taher[7], Shadi Hamouri[6,7]

1 Faculty of Medicine, Yarmouk University, Irbid, Jordan, 2 Faculty of Medicine, Mutah University, Al Karak, Jordan, 3 Faculty of Educational Sciences, Hashemite University, Al Zarqa, Jordan, 4 Faculty of Medicine, Hashemite University, Al Zarqa, Jordan, 5 School of Medicine, The University of Jordan, Amman, Jordan, 6 Faculty of Medicine, Jordan University of Science and Technology, Al-Ramtha, Jordan, 7 Faculty of Medicine, Al Balqa Applied University, Al Salt, Jordan

* Soha.beitawi@yu.edu.jo

## Abstract

### Background

Clinical preceptors serve as vital educators, so it is essential to enhance their effectiveness by developing a competency-based development program. In this study, we explored the challenges faced by preceptors and students, and measured the educational needs of preceptors, to inform the design of a syllabus for a preceptor development program.

### Methods

This was a sequential multi method study utilizing a structured questionnaire survey and focus group discussions among a representative sample of medical students in their fourth, fifth, and sixth years in addition to preceptors from the six public medical schools in Jordan.

### Results

Thematic analysis of focus group discussions revealed six themes: admission policy, training environment, curriculum gaps, trainers and mentorship, learners, and dissemination. The most important training needs documented by preceptors were teaching in the clinical setting, mentoring skills, simulation, assessment in the clinical setting, and providing feedback. Accordingly, a competency-based preliminary syllabus was developed.

**Data availability statement:** Data are provided as a supplementary file.

**Funding:** Funded by the European Union. Views and Opinions expressed are however those of the authors(s) only and do not necessary reflect those of the European Union or EACEA. Neither the European Union nor EACEA can be held responsible for them. Grant number: 101128823 The publication is based on data generated from a project. All project activities were funded by the European Union. The funder has no role in preparing the manuscript or decision to publish.

**Competing interests:** The authors declare no conflict of interest.

## Conclusion

It is essential to enrich the skills of preceptors regularly based on a needs assessment. Further long term studies are required to investigate the effectiveness of the proposed syllabus after implementation.

---

## Introduction

Preceptors are vital educators, guiding learners in real-world clinical settings. Faculty development programs (FDPs) enhance the teaching effectiveness of community-based preceptors through targeted training [1]. Community-based clinical education, integral to healthcare professionals' training, exposes learners to diverse patients and practice settings [2,3]. However, community preceptors often lack formal teaching training and struggle to balance clinical and educational roles [1,4]. FDPs address these challenges by equipping preceptors with the knowledge, skills, and support needed to excel in their educational roles.

In Jordan, medical education is provided over 6 years, the first 3 years includes basic sciences followed by three years of clinical sciences in the hospitals and health care centers from different sectors, universities, ministry of health and military medical services. It involves attending clinics, operation theatres, emergency rooms in addition to in patient wards. Clinical teaching is provided by academic staff from the faculties in addition to preceptors from community hospitals (ministry of health and military medical services).

Conducting a thorough needs assessment is essential for designing an effective FDP [1]. This process should account for the unique characteristics of community-based clinical settings, including variability in patient populations, resources, and organizational structures [5, 6]. Methods such as surveys, focus groups, interviews, and observation of preceptor-learner interactions can be employed towards that end [1,6].

Focus groups, in particular, provide a platform for clinical preceptors to share diverse perspectives, identify common challenges, and brainstorm solutions through guided discussions [7,8]. This method enhances the validity and relevance of findings, enabling stakeholders to design targeted training interventions tailored to preceptors' specific needs.

Structured questionnaires provide a standardized method to assess preceptors' knowledge, skills, and attitudes, ensuring consistent data collection. They offer a comprehensive evaluation of teaching methodologies, communication, assessment techniques, and inter-professional collaboration, identifying key training needs. Their standardized format promotes objective measurement, minimizing bias and ensuring empirical evidence guides training interventions. The resulting quantitative data support statistical analysis, prioritizing training initiatives for efficient resource allocation. Additionally, standardized questionnaires allow comparisons across preceptor cohorts or institutions, facilitating benchmarking and best practice identification to enhance quality and collaboration. They also serve as a feedback mechanism,

enabling preceptors to reflect on their strengths and weaknesses, fostering professional development and ongoing learning [9,10].

Effective FDPs should be tailored to community preceptors' identified needs and grounded in adult learning theory [11]. They may include workshops, seminars, online modules, mentorship, and ongoing support [3,12]. Programs should also incorporate strategies to engage preceptors and encourage participation in educational activities [5].

This study aimed to assess the teaching and mentoring needs of clinical preceptors (academic and community preceptors) in Jordanian medical schools, identify challenges in clinical training, and develop a preliminary syllabus for a preceptor training program based on these findings.

## Methods

### Study design

This is a cross-sectional sequential multi method study, that integrated a qualitative and quantitative approach to develop a preliminary syllabus for a preceptor development program using focus groups and a structured questionnaire. It comprised of three sequential phases:

(1) Exploratory phase: focus groups to identify challenges in clinical training for medical students and preceptors

(2) Quantitative phase: a structured national survey to assess preceptors' training needs

(3) Integration and development phase: synthesis of data from both phases by an expert panel to design a preliminary syllabus for a preceptor development program.

### Study population

The study included clinical preceptors (academic faculty and community hospitals: Ministry of Health hospitals, Military services Hospitals and private hospitals) who teach medical students in the six national medical schools in Jordan, along with medical students in the fourth, fifth, and sixth clinical years.

### Data collection tools and procedure

**Phase 1: Exploratory qualitative phase: focus groups discussion.** A stratified sample of 90 students (15 per medical school, balanced across clinical years, gender, and academic performance) and 60 preceptors (10 per medical school, representing all clinical specialties and affiliated hospitals) participated in focus groups, determined by expected thematic saturation. Additionally, 400 preceptors (academic and Ministry of Health staff) across the six medical schools completed the questionnaire to ensure a broad representation.

Focus groups were conducted with students and preceptors from six national Jordanian medical schools, within the time period 22nd December 2024–9th January 2025. Verbal Consent was obtained from all participants.

Students focus groups: each group included 15 students per university, tasked with identifying general challenges in clinical training that hinder learning in hospital settings, specific issues with clinical instructors, and perceived differences in teaching abilities between academic faculty and other instructors. Students also proposed solutions to enhance their training experience.

Preceptor focus groups: Each group included 10 preceptors per university, balanced across specialties. Preceptors identified challenges in training students, outlined essential skills and attributes of effective clinical trainers, and suggested training needs for a development program (Appendix 1: focus group form). A structured questionnaire, designed to assess preceptor training needs and complement focus group findings, was distributed via Google Forms to the clinical preceptors (academic faculty and Ministry of Health staff) across the six medical schools. (https://forms.office.com/r/g2AWdvAnXy).

**Phase 2: Quantitative phase: structured national survey.** The questionnaire was adapted from previously published work by Mukhalalati et al. [8]. Simple modification was done based on the focus groups results and the scope of this research. The questionnaire was validated by experts in medical education and tested for reliability on ten sample preceptors. The questionnaire included five sections, preceded by an agreement to participate:

1. Prioritizing training competencies (Lecturing for large groups; tutorial for small groups; bedside teaching, i.e., learning in the clinical setting; providing feedback; assessment in clinical setting; clinical simulation; and mentoring skills). Clinical simulation refer sto the ability of the preceptor to deploy a simulation scenario as an educating activity.

Prioritization involves choosing one of the following options: not a priority, low priority, medium priority, and high priority.

2. Motivations for joining a preceptor development program (multiple-choice)

3. Preferred training days and times (multiple-choice)

4. Preferred professional development methods (workshops and online modules)

5. Demographics (gender, specialty according to core board certification, years of experience, and health sector) and an open-ended comment section (see Appendix 2)

Dissemination: The questionnaire was distributed during the period 12th February 2024 up to 4th March 2024 via Google Forms to all clinical preceptors affiliated with Jordan's six national medical schools and associated Ministry of Health hospitals, which were identified through administrative lists. The lists included a group of preceptors and academics from each institute, in average 100–125 Academic and preceptor.

## Statistical analysis

The qualitative data were expressed in frequency and percentages; Chi-square of independence was used to explore association between categorical data particularly prioritizing of competency and preceptors' place of working. Bonferroni adjustment was used for multiple pairwise comparisons, P-value less than alpha level (0.05) deemed statistically significance. The data were analyzed using IBM SPSS Statistics (Version 28).

## Focus group data thematic analysis

Focus group transcripts were analyzed using a thematic analysis framework. Two researchers independently coded the data, resolving discrepancies through discussion. Thematic analysis, as outlined by Braun and Clarke, was used to analyze the focus group transcripts. This approach involved a six-phase process: familiarization with the data, generation of initial codes, searching for themes, reviewing and refining themes, defining and naming themes, and producing the final analysis. This method enabled the identification of recurring patterns and key themes related to the training needs of clinical instructors [13]. Six themes were identified, including challenges in clinical teaching and preceptor training needs, analyzed separately for student and preceptor groups to capture distinct perspectives. These themes were helpful in designing of the training program syllabus needed.

**Phase 3: Integration and syllabus development.** A panel of medical educators designed the preliminary syllabus, synthesizing focus group themes and questionnaire results. The syllabus included workshops on effective feedback and clinical simulation, addressing identified challenges and training needs.

## Ethical approval

Ethical approval was obtained from the Ethical Research Committee at Yarmouk University (IRB/2024/577). Participants provided informed consent, including verbal consent for focus groups and written consent for questionnaires.

Questionnaire respondents consented via a Google Forms consent form. Participants were not explicitly asked to consent to publication of their data; however, focus group transcripts were fully anonymized to ensure no potentially identifying information is included, as approved by the IRB. Data were anonymized and securely stored, with access restricted to authorized researchers.

## Results

### I. Focus Groups Results

A total of 90 students (15 per university) and 60 preceptors (10 per university) participated in focus groups across Jordan's six national medical schools. Thematic analysis identified six themes related to challenges in clinical training:

1. **Admission policy:** High student numbers due to open admission policies led to overcrowded clinical settings, reducing opportunities for hands-on practice.

2. **Training environment:** Variations in healthcare facilities (resource availability), limited access to emergency and critical care settings, and cultural barriers (restrictions on opposite-sex physical examinations) hindered effective training.

3. **Training preliminary syllabus**: Three subthemes emerged, highlighting preceptor-related challenges:

4. **Instructional materials**: Redundant content, lack of theme-based schedules, and insufficient coverage of topics like research skills, ethics, artificial intelligence, quality management, disaster medicine, and environmental medicine. Bedside teaching was often replaced by informal corridor discussions.

5. **Assessment**: Lack of standardized evaluation criteria and outdated clinical assessment methods.

6. **Feedback**: Absence of constructive feedback due to preceptors' lack of training in feedback delivery.

7. **Training and mentorship:** Preceptors struggled to balance clinical duties with teaching, establish safe learning environments, and communicate effectively. Variability in preceptors' teaching skills was noted.

8. **Learners:** Students reported low motivation, disconnection from preceptors, gaps between theoretical and clinical learning, and inadequate psychomotor skills.

9. **Program clarity:** Both groups highlighted unclear program learning outcomes and assessment methods, reducing training effectiveness.

### II. Training needs assessment

There were 400 respondents to the needs questionnaire. Table 1 demonstrates the general characteristics of the respondents (gender, specialty, years of expertise as a clinical educator, and health sector currently working in) while table 2 demonstrates the priority of training needs. When comparing training needs prioritizing between preceptors from University hospitals versus community hospitals (MOH, military, private), there was a statistically significant difference in lecturing for large group and assessment in the clinical setting. Where more preceptors from university hospital rated lecturing for large group as high priority (30.9% vs 18.3%)) while more community preceptors rate it as moderate priority (39.4% vs 20.8%), ($x^2 = 17.869$, df = 3, $P < 0.001$). On the other hands, more community preceptors rated assessment in the clinical setting as medium priority (34.7% vs 19.5%) while more preceptors from university hospitals rated as low priority (12.8% vs 5.2%), ($x^2 = 18.206$, $P < 0.001$). There was no statistically significant difference in other competencies.

Respondents were asked about what would influence them to register in the development program. The more frequently chosen factors were the objectives of training program and location, followed by providing a training certificate and timing of training sessions, Fig 1.

**Table 1. General characteristics of respondents.**

| Gender | # | % |
|---|---|---|
| Male | 271 | 67.75% |
| Female | 129 | 32.25% |
| TOTAL | 400 | 100.00% |
| **Specialty according to core board certification** | **#** | **%** |
| Internal medicine | 45 | 11.25% |
| Obstetrics and gynecology | 82 | 20.50% |
| Pediatrics | 74 | 18.50% |
| Surgery | 65 | 16.25% |
| Other | 134 | 33.50% |
| TOTAL | 400 | 100.00% |
| **Years of expertise as a clinical educator** | **#** | **%** |
| 1-5 | 131 | 32.75% |
| 6-10 | 80 | 20.00% |
| 11-15 | 73 | 18.25% |
| >15 | 116 | 29.00% |
| TOTAL | 400 | 100.00% |
| **Health sector currently working in** | **#** | **%** |
| Military services | 61 | 15.25% |
| Ministry of Health | 156 | 39.00% |
| Private sector | 34 | 8.50% |
| University Hospital | 149 | 37.25% |
| TOTAL | 400 | 100.00% |

**Training program delivery preferences.** Questionnaire results (n = 400 preceptors) informed the training program's delivery model and schedule (Table 3). Preceptors preferred face-to-face training (42.8%), followed by blended (36.5%) and online (20.8%) methods. Morning sessions (10:00 am–12:00 noon: 34.7%; 8:00 am–10:00 am: 21.1%) on Sunday (19.3%), Tuesday (18.0%), or Monday (17.5%) were most favored, reflecting work schedule constraints. These preferences shaped the hybrid delivery model (online resources and face-to-face sessions) and morning scheduling of the training modules.

### III. Suggested content and learning objectives of training program for clinical preceptors

Based on focus group themes (lack of feedback, inconsistent assessments) and questionnaire results (high-priority competencies; Table 3), the training program addresses five core competencies: bedside teaching, mentorship, clinical simulation, assessment, and feedback. The program adopts a hybrid delivery model, with online resources (e.g., recorded lectures) and face-to-face interactive sessions held in the morning on working days, as preferred by preceptors (Table 3). Feasibility considerations, such as preceptor availability and institutional resources, informed the program's design.

The training program focuses on five high-priority competencies identified by preceptors (Table 3), addressing key challenges from focus groups. Small group tutorials and lecturing were excluded due to lower prioritization.

### a- Bedside Teaching

Preceptors will master strategies for engaging students in clinical settings, setting clear learning objectives, and integrating theoretical and practical skills. They will learn to promote critical thinking and evidence-based decision-making in a supportive environment.

**Table 2. Training needs priority.**

| Lecturing for large groups | # | % |
| --- | --- | --- |
| High priority | 92 | 23.00% |
| Medium priority | 130 | 32.50% |
| Low priority | 103 | 25.75% |
| Not a priority | 75 | 18.75% |
| TOTAL | 400 | 100.00% |
| **Tutorial for small groups** | **#** | **%** |
| High priority | 187 | 46.75% |
| Medium priority | 127 | 31.75% |
| Low priority | 49 | 12.25% |
| Not a priority | 37 | 9.25% |
| TOTAL | 400 | 100.00% |
| **Bedside teaching (learning in a clinical setting)** | **#** | **%** |
| High priority | 249 | 62.25% |
| Medium priority | 67 | 16.75% |
| Low priority | 47 | 11.75% |
| Not a priority | 37 | 9.25% |
| TOTAL | 400 | 100.00% |
| **Providing feedback** | **#** | **%** |
| High priority | 198 | 49.50% |
| Medium priority | 130 | 32.50% |
| Low priority | 48 | 12.00% |
| Not a priority | 24 | 6.00% |
| TOTAL | 400 | 100.00% |
| **Assessment in clinical setting** | **#** | **%** |
| High priority | 223 | 55.75% |
| Medium priority | 116 | 29.00% |
| Low priority | 32 | 8.00% |
| Not a priority | 29 | 7.25% |
| TOTAL | 400 | 100.00% |
| **Clinical simulation** | **#** | **%** |
| High priority | 230 | 57.50% |
| Medium priority | 104 | 26.00% |
| Low priority | 42 | 10.50% |
| TOTAL | 400 | 100.00% |
| **Mentoring skills** | **#** | **%** |
| High priority | 233 | 58.25% |
| Medium priority | 120 | 30.00% |
| Low priority | 25 | 6.25% |
| Not a priority | 22 | 5.50% |
| TOTAL | 400 | 100.00% |

b- **Mentorship**

Preceptors will develop skills to build effective mentor-mentee relationships, foster trust, and address challenges (e.g., resolving conflicts). They will apply evidence-based practices to promote diversity, equity, and inclusion in mentorship.

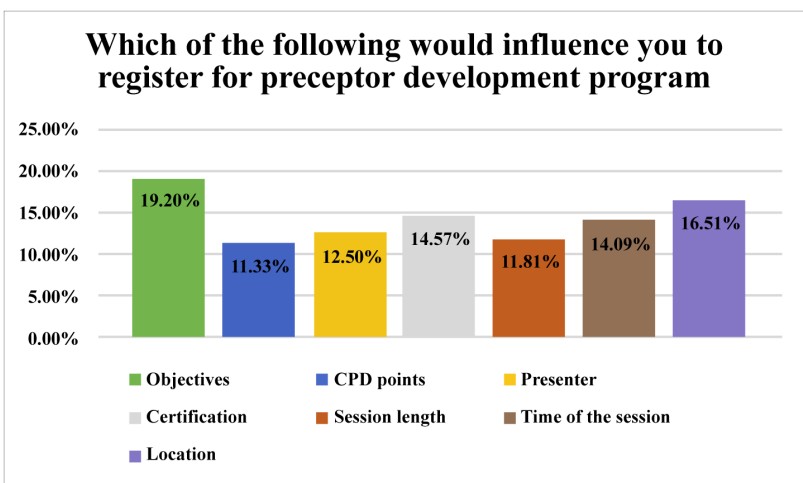

**Fig 1. Factors that would influence attending the development program.**

**Table 3. Preceptor-reported high-priority training competencies.**

| Competency | Percentage rating high priority need |
| --- | --- |
| Bedside teaching | 62% |
| Mentoring skills | 59% |
| Clinical simulation | 58% |
| Assessment in the clinical setting | 56% |
| Providing feedback | 50% |
| Small group tutorial | 46% |
| Lecturing large groups | 23% |

### c- Clinical Simulation

Preceptors will learn to deploy simulation scenarios, conduct pre-briefing and debriefing, and evaluate learners and instructors, integrating simulations into the curriculum.

### d- Assessment in the clinical setting

Preceptors will apply diverse assessment methods (direct observation, standardized patients, case-based assessments, and written examinations) to evaluate student competence accurately and inform curricular improvements.

### e- Feedback

Preceptors will develop skills to deliver constructive, tailored feedback, engage students in reflective discussions to promote self-reflection, goal setting, and active participation in the feedback process, and foster a culture of continuous improvement.

The training program is structured into four modules, covering the five high-priority competencies (Table 3). Each module spans 3–4 days (4 hours daily) and is delivered via a hybrid model (online lectures and resources, face-to-face seminars, and case-based tabletop exercises) in the morning from Sunday to Tuesday, as preferred by preceptors (Table 3). Program effectiveness will be evaluated through participant feedback and skill assessments.

Module 1: **Introduction to preceptorship**: Covers core responsibilities, ethical standards, and mentorship skills, including building trust and promoting diversity in mentor-mentee relationships.

Module 2: **Effective communication**: Focuses on understanding feedback principles, delivering constructive feedback, and engaging students in reflective discussions.

Module 3: **Teaching in the clinical setting**: Addresses bedside teaching and clinical simulation, including planning sessions, promoting critical thinking, and designing simulation scenarios with debriefing.

Module 4: **Assessment and evaluation**: Covers clinical assessment methods (e.g., direct observation, standardized patients) and strategies for evaluating student competence and program outcomes.

## Discussion

### Defining challenges and needs assessment

This study aimed to design a preliminary syllabus for a clinical preceptor training program by identifying the challenges in medical education and assessing preceptor training needs in Jordan. The findings highlight actionable challenges and needs, informing a tailored faculty development program.

### Challenges in clinical training

Focus groups revealed significant barriers to effective clinical training. Over-enrollment led to overcrowded clinical settings and limited hands-on practice, as reported by both students and preceptors. This underscores the need for policy changes, such as stricter admission criteria by the Ministry of Higher Education, to align student numbers with available resources. Cultural barriers, particularly restrictions on opposite-sex physical examinations, hindered consistent competency development. These could be addressed through simulation-based training, as included in the proposed syllabus, to provide standardized learning opportunities [11]. Curriculum gaps, including insufficient coverage of topics like research skills, ethics, and disaster medicine, highlight the need for curricular innovation to prepare students for modern healthcare demands. Pedagogical deficiencies, such as replacing bedside teaching with informal discussions, further necessitate preceptor training to enhance teaching skills [11].

### Preceptor training needs

Preceptors' lack of formal training in medical education, as noted in the International Association of the Health Profession Education (AMEE) Guide 52 [11], was evident in reported deficiencies in feedback delivery, clinical assessment, and bedside teaching. The mixed method approach aligns with established methods for assessing learning needs which emphasize integrating qualitative and quantitative data to ensure relevance [12]. The identified priorities suggest preceptors are aware of their skill gaps, indicating potential motivation for training that can bridge the gap between current abilities and effective clinical teaching [7,12]. Our findings coincide with Mukhalati et. al. where teaching in the clinical setting, mentoring, assessment in the clinical setting were identified as high priority needs. In addition to educational research skills but we didn't ask about it [8].

Time constraints, a known challenge for clinical teaching [11], were considered by aligning the program with preceptors' preferences for morning sessions on working days (Sunday-Tuesday) and face-to-face delivery. The hybrid model, combining online resources and face-to-face seminars, enhances feasibility by accommodating preceptors' clinical schedules, as recommended for effective faculty development [11]. These practical considerations ensure the program is accessible and relevant in Jordan's resource-constrained context.

### Developing the syllabus

Competency-based education is becoming a core strategy worldwide and is driven by the perceived needs. The competency of preceptors will help enhance the competency of students [14,15]. Knowing the high priority training competencies

as reported by the preceptor allow designing a focused syllabus. The identified five competencies are categorized into 4 modules, which will be delivered in the preferred reported method (hybrid) by the preceptors

### Rationale for competencies

Bedside teaching, the highest-priority competency, reflects preceptors' need for training to deliver structured clinical instruction, addressing focus group reports of informal discussions replacing bedside teaching. Mentoring training tackles students' reported disconnection from preceptors, which can hinder development and goal achievement if ineffective [16]. Proper mentorship training fosters supportive relationships, enhancing learning outcomes. Clinical simulation, prioritized to address cultural barriers (e.g., restrictions on opposite-sex examinations) and limited emergency care exposure, allows safe skill practice while controlling task consequences, aligning with its role in modern medical education [17]. Accordingly, preceptors should be oriented and trained in utilizing simulation efficiently. Assessment training responds to focus group findings of non-standardized evaluation criteria, enabling preceptors to evaluate student competence accurately, shaping educational quality and student behavior [18]. Feedback training addresses the reported lack of constructive feedback, a critical tool for improving student performance and fostering continuous learning [19]. These competencies aim to elevate clinical training and healthcare delivery.

This study is characterized by implementing two methods, focus groups and a structured questionnaire. The mixed method approach, combining focus groups and questionnaires, captured unanticipated challenges, enhancing the syllabus's relevance [20]. Moreover, interviews can provide in depth insight into participant's perspective and allow explanation of information [7].

### Strengths and limitations

The study's strength lies in its comprehensive needs assessment, involving 90 students and 400 preceptors from six Jordanian medical schools, ensuring diverse representation across specialties and experience levels. This large sample may encourage widespread preceptor engagement in the training program. The mixed method design provided in-depth insights, with focus groups revealing unanticipated issues that complemented questionnaire data [19]. However, reliance on perceptions may introduce bias, and the focus on Jordanian medical schools' limits generalizability. Objective assessment of educational outcomes post-implementation is needed to validate the program's effectiveness.

### Conclusion

This study identified critical teaching and mentoring needs among clinical preceptors in Jordanian medical schools, informing a competency-based syllabus for a preceptor training program. After addressing challenges like over-enrollment and feedback deficiencies, the program aims to enhance medical education quality. Objective evaluation of its impact on preceptor performance and student outcomes is essential, with regular updates to ensure alignment with evolving medical and educational demands. Stakeholder engagement, including the Ministry of Higher Education, will support sustainable implementation and broader policy reforms.

### Supporting information

**S1 File. Needs Assessment Structured Questionnaire.**
(DOCX)

**S1 Table. Original Data for all universities.**
(XLSX)

## Acknowledgments

We would like to express our sincere gratitude to Professor Manar Al-lawama for her invaluable contribution to this study. Her support in designing the questionnaire and focus group format, in addition to her assistance in drafting the manuscript, was essential to the completion of this work.

## Author contributions

**Conceptualization:** Randa Mahasneh.

**Data curation:** Emad Aborajooh, Walid I Wadi, Mohammad Alqudah, Fida Asali, Ahmed Sheyyab, Motasem Al-latayfeh, Enas Al-Zayadneh, Eman Badran, Kais Al balbissi, Raed Al-Taher, Asma Basha, Rola Saqan, Ashraf Omar Oweis, Wafa Taher.

**Funding acquisition:** Randa Mahasneh.

**Methodology:** Soha Albeitawi, Anas Aljaiuossi, Murad Shatnawi, Enas Al-Zayadneh, Yaser M. Rayyan.

**Project administration:** Randa Mahasneh.

**Supervision:** Soha Albeitawi, Ahlam Al-Kharabsheh, Fadi Sawaqed, Rola Saqan.

**Validation:** Eman Badran.

**Visualization:** Yaser M. Rayyan, Ashraf Omar Oweis, Wafa Taher, Shadi Hamouri.

**Writing – original draft:** Soha Albeitawi, Anas Aljaiuossi, Murad Shatnawi, Ahlam Al-Kharabsheh, Fida Asali, Ahmed Sheyyab, Enas Al-Zayadneh, Wafa Taher.

**Writing – review & editing:** Soha Albeitawi, Mohammad Talal Al-Zubi, Fadi Sawaqed, Emad Aborajooh, Walid I Wadi, Randa Mahasneh, Benjamin Rowland Colton, Mohammad Alqudah, Tamara Kufoof, Monther A. Gharaibeh, Motasem Al-latayfeh, Eman Badran, Yaser M. Rayyan, Kais Al balbissi, Raed Al-Taher, Asma Basha, Rola Saqan, Ashraf Omar Oweis, Shadi Hamouri.

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
