## [Decision Letter · Decision Letter 0]

22 Jul 2025

Dear Dr. Albeitawi,

**I would specifically advise you to restructure the paper for improve its coherence and  attach the survey questionnaire as an appendix for transparency and reproducibility.**

We look forward to receiving your revised manuscript.

Kind regards,

Mariam Shadan, MBBS, MSc, MD

Academic Editor

PLOS ONE

**Journal requirements:**

1. When submitting your revision, we need you to address these additional requirements. Please ensure that your manuscript meets PLOS ONE's style requirements, including those for file naming. The PLOS ONE style templates can be found at https://journals.plos.org/plosone/s/file?id=wjVg/PLOSOne_formatting_sample_main_body.pdf and https://journals.plos.org/plosone/s/file?id=ba62/PLOSOne_formatting_sample_title_authors_affiliations.pdf 2. We note that the grant information you provided in the ‘Funding Information’ and ‘Financial Disclosure’ sections do not match.  When you resubmit, please ensure that you provide the correct grant numbers for the awards you received for your study in the ‘Funding Information’ section. 3. Thank you for stating the following financial disclosure: Funded by the European Union. Views and Opinions expressed are however those of the authors(s) only and do not necessary reflect those of the European Union or EACEA. Neither the European Union nor EACEA can be held responsible for them.   Please state what role the funders took in the study.  If the funders had no role, please state: "The funders had no role in study design, data collection and analysis, decision to publish, or preparation of the manuscript." If this statement is not correct you must amend it as needed. Please include this amended Role of Funder statement in your cover letter; we will change the online submission form on your behalf. 4. In the online submission form, you indicated that “Data are available from the corresponding author upon reasonable request”.  All PLOS journals now require all data underlying the findings described in their manuscript to be freely available to other researchers, either a. In a public repository, b. Within the manuscript itself, or c. Uploaded as supplementary information.This policy applies to all data except where public deposition would breach compliance with the protocol approved by your research ethics board. If your data cannot be made publicly available for ethical or legal reasons (e.g., public availability would compromise patient privacy), please explain your reasons on resubmission and your exemption request will be escalated for approval. 5. We note that this data set consists of interview transcripts. Can you please confirm that all participants gave consent for interview transcript to be published? If they DID provide consent for these transcripts to be published, please also confirm that the transcripts do not contain any potentially identifying information (or let us know if the participants consented to having their personal details published and made publicly available). We consider the following details to be identifying information:- Names, nicknames, and initials- Age more specific than round numbers- GPS coordinates, physical addresses, IP addresses, email addresses- Information in small sample sizes (e.g. 40 students from X class in X year at X university)- Specific dates (e.g. visit dates, interview dates)- ID numbers Or, if the participants DID NOT provide consent for these transcripts to be published:- Provide a de-identified version of the data or excerpts of interview responses- Provide information regarding how these transcripts can be accessed by researchers who meet the criteria for access to confidential data, including:a) the grounds for restrictionb) the name of the ethics committee, Institutional Review Board, or third-party organization that is imposing sharing restrictions on the datac) a non-author, institutional point of contact that is able to field data access queries, in the interest of maintaining long-term data accessibility.d) Any relevant data set names, URLs, DOIs, etc. that an independent researcher would need in order to request your minimal data set. For further information on sharing data that contains sensitive participant information, please see: https://journals.plos.org/plosone/s/data-availability#loc-human-research-participant-data-and-other-sensitive-data If there are ethical, legal, or third-party restrictions upon your dataset, you must provide all of the following details (https://journals.plos.org/plosone/s/data-availability#loc-acceptable-data-access-restrictions):a) A complete description of the datasetb) The nature of the restrictions upon the data (ethical, legal, or owned by a third party) and the reasoning behind themc) The full name of the body imposing the restrictions upon your dataset (ethics committee, institution, data access committee, etc)d) If the data are owned by a third party, confirmation of whether the authors received any special privileges in accessing the data that other researchers would not havee) Direct, non-author contact information (preferably email) for the body imposing the restrictions upon the data, to which data access requests can be sent  

Reviewers' comments:

Reviewer's Responses to Questions

**Comments to the Author**

1. Is the manuscript technically sound, and do the data support the conclusions?

Reviewer #1: Yes

Reviewer #2: Partly

2. Has the statistical analysis been performed appropriately and rigorously?

Reviewer #1: N/A

Reviewer #2: N/A

3. Have the authors made all data underlying the findings in their manuscript fully available?

Reviewer #1: Yes

Reviewer #2: No

4. Is the manuscript presented in an intelligible fashion and written in standard English?

Reviewer #1: Yes

Reviewer #2: Yes

**Reviewer #1:**  This paper provides insights into variables that may impact clinical preceptor effectiveness in teaching. While limited to one country, the study had the advantage of studying six medical schools with significant numbers of preceptor and student participants to make the conclusions justified. I did not see that a copy of the written survey was readily available and readers may find this to be a useful appendix if they desire to investigate their own institution. Perhaps it was included and I somehow glossed over it in my reading of the manuscript.

The term 'clinical simulation' is intuitively obvious but I think a few sentences to describe what they mean might be helpful as a guide to this training need priority.

Table 3 outlining preferences for preceptor day and time for participation takes up a lot of space and could be reduced to a few sentences. The findings are an important variable but may likely differ among institutions/countries.

**Reviewer #2:**  Overall, this paper represents an important attempt to define the gaps in clinical teaching in medical schools in Jordan, to inform preceptor training. The study design is appropriate, with use of both qualitative data from a survey instrument, and focus groups. The sample size is impressive.

The paper would benefit from some reorganization, so that the study part is clearly separated from the resulting curriculum development part. The analysis is limited to descriptive frequencies, which seems like a missed opportunity to me. With the range of respondents representing students, as well as faculty in two different settings, surely some meaningful comparisons could have been made. Did all the groups agree in their responses? I imagine that analysis of the student responses in the focus groups would yield some important differences from the responses of the faculty. If the focus groups were mixed, with students and preceptors together (which needs clarification), then would your participant coding permit any analysis by type of participant? I also think the work setting of the preceptors might have led to some differences in identified challenges and needs addressed in both the survey and the focus groups. With 400 survey respondents, you should be able to perform some analysis of respondent differences by health sector.

Another problem for me as a reader is lack of familiarity with the Jordanian medical education system. An introduction of the system would be helpful to readers who may make assumptions based on the system they know. This could perhaps be provided in a graphical timeline format.

I will now provide detailed line-by-line comments and suggestions.

1. The word “Based” in the title is unnecessary, and the flow suggests that the assessment may also include assessment of the design of the curriculum. I recommend changing the title to: “Assessment of Clinical Medical Education Needs to Inform Design of a Preceptor Development Program in Jordan: A National Mixed Method Study.”

2. The funding details statement appears to be missing some important required details.

3. The data availability statement does not appear to meet the PLOS One requirements.

4. The abstract appears in two versions, the first starting on Page 2, line 32 and ending on page 3, line 58, and the second version starting on page 3, line 59 and ending on page 4, line 79. Even the key words are different. Please decide which abstract content you want to keep as your final.

5. Page 3, lines 61-63 in Background: “In this study,…” could be reworded for clarity to: “In this study, we explored the challenges faced by preceptors and students, and measured the educational needs of preceptors, to inform the design of a syllabus for a preceptor development program.”

6. Page 4, line 65, the words “questionnaire” and “survey” are redundant. Please choose one.

7. The introduction needs an overview of the medical education system in Jordan, including the time spent by students (do students enter medical school after college, or do the early years of medical school serve as college? Is the education divided into preclinical (classroom) training and then clinical training? Do the 6 years of medical school education include internship/residency? Are there required rotations and electives? Who are the preceptors? There is mention of some being community and some being academic and some Ministry of Health. In my country, medical students in their 3rd and 4th year have required inpatient rotations, mostly at the main teaching hospital, but also have outpatient rotations (some with academic faculty and some with community preceptors), and can take rotations at community hospitals. As you may imagine, the teaching skills and approaches vary immensely. This paper needs a better description of the preceptors, their sites/settings and their representation as teachers of the students. (If most of the precepting is done by community preceptors but most of your respondents were academic faculty, this would be an important weakness.)

8. On page 6 and 7, study design, I would like to see more granularity about the participants and their work settings. On lines 124-126, you state that “The study included clinical preceptors (academic faculty and Ministry of Health hospital staff)…” But elsewhere you refer to community preceptors (introduction, lines 83-86)? On page 7, lines 139-140, you refer to clinical instructors? Who are they? Are all the preceptors included in this study teaching in inpatient settings? Perhaps a graphic or at least a paragraph describing the preceptor types and their work settings is needed.

9. Page 8 line 149, I think you mean “adapted” rather than “adopted”.

10. On page 8, line 153, I recommend clarifying that the prioritization used a Likert type scale (rather than a ranking). In the same line, it would improve reading comprehension to separate the training competency topics with semicolons.

11. Page 8 line 161, I recommend change “among” to “during” and delete “up” In the date range.

12. The dissemination should be better explained. How many preceptors were on the list? How many responded? Any response bias based on practice setting?

13. It would be helpful to know whether there were any research incentives for either the students or the preceptors.

14. Page 9 Lines 166-171, I would like more information about the approach to thematic analysis. Did you use or adapt any specific approach to coding the focus group themes? If so, please explain and provide a reference.

15. Page 9, lines 160-171, this sentence is confusing: “These themes made us aware of the modification of the questionnaire and the design of the training program syllabus needed.” Do you mean: “These themes informed needed revisions in the questionnaire and the eventual design of the training program syllabus.”?

16. Page 9 lines 172-175 seems out of place and the section title is confusing. I think this belongs at the end (it is not really part of your study) as it describes the approach to designing the curriculum based on your study results. I recommend changing the header to “Syllabus Design Process”.

17. The results section would benefit from more comparative analysis, as addressed in my introductory paragraphs.

18. On page 14, line 218, please clarify what you mean by “certification,” In my world, this has a very specific meaning, based on taking a board exam following specialty training. From context here, I believe you mean providing a training certificate.

19. Page 15, lines 226-227 – description of the eventual curriculum development does not belong in results.

20. Page 17, line 251, do you really mean that your training program would prepare preceptors to design simulation scenarios? This seems more ambitious than you can likely accomplish in the amount of time you have planned, and is likely beyond reach of many clinical preceptors. Perhaps you mean that they would be expected to use, adapt, deploy simulation scenarios that are provided to them?

21. Page 18, line 229, I recommend deleting the word “reflective”, which is redundant with “self-reflection a few words later.

22. Page 19, lines 19-282, I recommend rewording this to describe the study, designed to inform the curriculum, as the study is not really about the curriculum.

23. I recommend reorganizing the discussion and shortening it. There is a great degree of redundancy. Discuss the results in reference to other related papers, then discuss how the results were applied in the planned curriculum.

24. There may be limitations based on selection bias of respondents, if the distribution of those responding did not reflect the range of preceptors. What is the risk that those who need the development course least are the most likely to take it (because they love to teach and are interested in it)?

25. There are citations to several papers missing from the reference list: 16-19.

26. Page 22 Conclusion. You state: “By addressing challenges like over-enrollment and feedback deficiencies, the program aims to enhance medical education quality.” How does your study or your curriculum plan address the problem of over enrollment?

**Do you want your identity to be public for this peer review?** For information about this choice, including consent withdrawal, please see our Privacy Policy

Reviewer #1: No

Reviewer #2: No

---

## [Author Response · Author response to Decision Letter 1]

5 Aug 2025

Thank you for submitting your manuscript to PLOS ONE. After careful consideration, we feel that it has merit but does not fully meet PLOS ONE’s publication criteria as it currently stands. Therefore, we invite you to submit a revised version of the manuscript that addresses the points raised during the review process.

I would specifically advise you to restructure the paper for improve its coherence and attach the survey questionnaire as an appendix for transparency and reproducibility.

thank you for this comment, the survey questionnaire had been already attached (submitted to system)

rebuttal letter is uploaded.

thank you, done as requested

statement is updated as requested

not applicable

We look forward to receiving your revised manuscript.

Kind regards,

Mariam Shadan, MBBS, MSc, MD

Academic Editor

PLOS ONE

thank you for this comment, done as requested

thank you, they should match now

thank you, number is added to the funding statement

Funded by the European Union. Views and Opinions expressed are however those of the authors(s) only and do not necessary reflect those of the European Union or EACEA. Neither the European Union nor EACEA can be held responsible for them.

The publication is based on data generated from a project. All project activities were funded by the European Union. The funder has no role in preparation of the manuscript.

included in the cover letter

4. In the online submission form, you indicated that “Data are available from the corresponding author upon reasonable request”.

data is provided as supplementary file

5. We note that this data set consists of interview transcripts. Can you please confirm that all participants gave consent for interview transcript to be published?

the provided interview transcript is the form used to fill in the interview however, it doesn't include data.

If they DID provide consent for these transcripts to be published, please also confirm that the transcripts do not contain any potentially identifying information (or let us know if the participants consented to having their personal details published and made publicly available). We consider the following details to be identifying information:

- Names, nicknames, and initials

- Age more specific than round numbers

- GPS coordinates, physical addresses, IP addresses, email addresses

- Information in small sample sizes (e.g. 40 students from X class in X year at X university)

- Specific dates (e.g. visit dates, interview dates)

- ID numbers

Or, if the participants DID NOT provide consent for these transcripts to be published:

- Provide a de-identified version of the data or excerpts of interview responses

- Provide information regarding how these transcripts can be accessed by researchers who meet the criteria for access to confidential data, including:

a) the grounds for restriction

b) the name of the ethics committee, Institutional Review Board, or third-party organization that is imposing sharing restrictions on the data

c) a non-author, institutional point of contact that is able to field data access queries, in the interest of maintaining long-term data accessibility.

d) Any relevant data set names, URLs, DOIs, etc. that an independent researcher would need in order to request your minimal data set.

For further information on sharing data that contains sensitive participant information, please see: https://journals.plos.org/plosone/s/data-availability#loc-human-research-participant-data-and-other-sensitive-data

If there are ethical, legal, or third-party restrictions upon your dataset, you must provide all of the following details (https://journals.plos.org/plosone/s/data-availability#loc-acceptable-data-access-restrictions):

a) A complete description of the dataset

b) The nature of the restrictions upon the data (ethical, legal, or owned by a third party) and the reasoning behind them

c) The full name of the body imposing the restrictions upon your dataset (ethics committee, institution, data access committee, etc)

d) If the data are owned by a third party, confirmation of whether the authors received any special privileges in accessing the data that other researchers would not have

e) Direct, non-author contact information (preferably email) for the body imposing the restrictions upon the data, to which data access requests can be sent

Reviewers' comments:

Reviewer's Responses to Questions

Comments to the Author

1. Is the manuscript technically sound, and do the data support the conclusions?

Reviewer #1: Yes

Reviewer #2: Partly

2. Has the statistical analysis been performed appropriately and rigorously?

Reviewer #1: N/A

Reviewer #2: N/A

3. Have the authors made all data underlying the findings in their manuscript fully available?

Reviewer #1: Yes

Reviewer #2: No

4. Is the manuscript presented in an intelligible fashion and written in standard English?

Reviewer #1: Yes

Reviewer #2: Yes

5. Review Comments to the Author

Reviewer #1: This paper provides insights into variables that may impact clinical preceptor effectiveness in teaching. While limited to one country, the study had the advantage of studying six medical schools with significant numbers of preceptor and student participants to make the conclusions justified. I did not see that a copy of the written survey was readily available and readers may find this to be a useful appendix if they desire to investigate their own institution. Perhaps it was included and I somehow glossed over it in my reading of the manuscript.

The term 'clinical simulation' is intuitively obvious but I think a few sentences to describe what they mean might be helpful as a guide to this training need priority.

Table 3 outlining preferences for preceptor day and time for participation takes up a lot of space and could be reduced to a few sentences. The findings are an important variable but may likely differ among institutions/countries.

Reply: thank you for your valuable comments. The Questionnaire and the interview topics outlines had been already submitted as appendix files.

Clinical simulation: means the need to be trained how to deploy simulation scenarios. Clarified in lines: 162-163

Table 3 is omitted and the most important findings has been already mentioned in lines: 238-242

Reviewer #2: Overall, this paper represents an important attempt to define the gaps in clinical teaching in medical schools in Jordan, to inform preceptor training. The study design is appropriate, with use of both qualitative data from a survey instrument, and focus groups. The sample size is impressive.

The paper would benefit from some reorganization, so that the study part is clearly separated from the resulting curriculum development part. The analysis is limited to descriptive frequencies, which seems like a missed opportunity to me. With the range of respondents representing students, as well as faculty in two different settings, surely some meaningful comparisons could have been made. Did all the groups agree in their responses? I imagine that analysis of the student responses in the focus groups would yield some important differences from the responses of the faculty. If the focus groups were mixed, with students and preceptors together (which needs clarification), then would your participant coding permit any analysis by type of participant? I also think the work setting of the preceptors might have led to some differences in identified challenges and needs addressed in both the survey and the focus groups. With 400 survey respondents, you should be able to perform some analysis of respondent differences by health sector.

Author reply: Thank you for this valuable comment, though comparative analysis won’t significantly affect the design to syllabus, we compared the responses regarding training needs priority between the preceptors from university hospitals versus community preceptors.

Another problem for me as a reader is lack of familiarity with the Jordanian medical education system. An introduction of the system would be helpful to readers who may make assumptions based on the system they know. This could perhaps be provided in a graphical timeline format.

Author reply: Thank you for forwarding attention to this. A paragraph describing the Jordanian medical education system is added into the introduction.

In Jordan, medical education is provided over 6 years, the first 3 years includes basic sciences followed by three years of clinical sciences in the hospitals and health care centers from different sectors, universities, ministry of health and military medical services. It involves attending clinics, operation theatres, emergency rooms in addition to in patient wards. Clinical teaching is provided by academic staff from the faculties in addition to community preceptors from ministry of health and military medical services.

I will now provide detailed line-by-line comments and suggestions.

1. The word “Based” in the title is unnecessary, and the flow suggests that the assessment may also include assessment of the design of the curriculum. I recommend changing the title to: “Assessment of Clinical Medical Education Needs to Inform Design of a Preceptor Development Program in Jordan: A National Mixed Method Study.”

Thank you for your comment. The title changed as requested

2. The funding details statement appears to be missing some important required details.

Thank you for this comment. Details about the funder’s role is added.

3. The data availability statement does not appear to meet the PLOS One requirements.

Thank you for your comment. Data is now provided as supplementary file.

4. The abstract appears in two versions, the first starting on Page 2, line 32 and ending on page 3, line 58, and the second version starting on page 3, line 59 and ending on page 4, line 79. Even the key words are different. Please decide which abstract content you want to keep as your final.

Thank you for this comment. The first one is omitted.

5. Page 3, lines 61-63 in Background: “In this study,…” could be reworded for clarity to: “In this study, we explored the challenges faced by preceptors and students, and measured the educational needs of preceptors, to inform the design of a syllabus for a preceptor development program.”

Thank you for this comment. Changed as requested.

6. Page 4, line 65, the words “questionnaire” and “survey” are redundant. Please choose one.

Thank you for this comment. The word survey is deleted.

7. The introduction needs an overview of the medical education system in Jordan, including the time spent by students (do students enter medical school after college, or do the early years of medical school serve as college? Is the education divided into preclinical (classroom) training and then clinical training? Do the 6 years of medical school education include internship/residency? Are there req

---

## [Decision Letter · Decision Letter 1]

27 Oct 2025

Dear Dr. Albeitawi,

Thank you for submitting your manuscript to PLOS ONE. After careful consideration, we feel that it has merit but does not fully meet PLOS ONE’s publication criteria as it currently stands. Therefore, we invite you to submit a revised version of the manuscript that addresses the points raised during the review process.

Thank you for responding to the reviewer comments appropriately.

1. Please remove the word "national" in front of the mixed-method study.

2. Secondly, since the final outcome of the study is from an expert panel who developed the syllabus. The results of preliminary studies, focus group discussions and surveys were sequential, with the focus group discussion providing data used to contextualise an existing questionnaire for the subsequent phase. Thus, the study is best characterised by a sequential, multimethod design. have a look at these exaples to guide you-https://pubmed.ncbi.nlm.nih.gov/32554723/,   https://books.google.co.za/books?hl=en&lr=&id=F8BFOM8DCKoC&oi=fnd&pg=PA189&dq=sequential+multimethod+design&ots=gXgMyBoxJg&sig=l1bh8zh6jyYBJazHH3WuuL_m72I#v=onepage&q=sequential%20multimethod%20design&f=false

Please rewrite the methodology session 

We look forward to receiving your revised manuscript.

Kind regards,

Christmal Dela Christmals, PhD, MSc, BSc, RN

Academic Editor

PLOS ONE

Journal Requirements:

Reviewers' comments:

Reviewer's Responses to Questions

**Comments to the Author**

Reviewer #1: All comments have been addressed

Reviewer #2: All comments have been addressed

2. Is the manuscript technically sound, and do the data support the conclusions?

Reviewer #1: Yes

Reviewer #2: Yes

3. Has the statistical analysis been performed appropriately and rigorously?

Reviewer #1: Yes

Reviewer #2: Yes

4. Have the authors made all data underlying the findings in their manuscript fully available?

Reviewer #1: Yes

Reviewer #2: Yes

5. Is the manuscript presented in an intelligible fashion and written in standard English?

Reviewer #1: Yes

Reviewer #2: Yes

Reviewer #1: (No Response)

Reviewer #2: Thank you for accepting my recommended changes and addressing my concerns. Good luck with your planned training program!

**Do you want your identity to be public for this peer review?** For information about this choice, including consent withdrawal, please see our Privacy Policy

Reviewer #1: No

Reviewer #2: No

---

## [Author Response · Author response to Decision Letter 2]

31 Oct 2025

Dear Editor

We appreciate the valuable comments you provided us with. We addressed the comments as requested and we hope by this it fully fits the PLOS one’s publication criteria.

Kindly find below response to each comment.

Editor/ Reviewers Comments:

Comment 1: Please remove the word "national" in front of the mixed-method study.

Reply: thank you for your comment. The word national is removed and mixed changed into multi

Comment 2: Secondly, since the final outcome of the study is from an expert panel who developed the syllabus. The results of preliminary studies, focus group discussions and surveys were sequential, with the focus group discussion providing data used to contextualise an existing questionnaire for the subsequent phase. Thus, the study is best characterised by a sequential, multimethod design. have a look at these examples to guide you-https://pubmed.ncbi.nlm.nih.gov/32554723/, https://books.google.co.za/books?hl=en&lr=&id=F8BFOM8DCKoC&oi=fnd&pg=PA189&dq=sequential+multimethod+design&ots=gXgMyBoxJg&sig=l1bh8zh6jyYBJazHH3WuuL_m72I#v=onepage&q=sequential%20multimethod%20design&f=false

Please rewrite the methodology session

Reply: thank you for your valuable comment, the method section is modified to a sequential multi method theme

---

## [Editor Report · Decision Letter 2]

4 Nov 2025

Assessment of Clinical Medical Education Needs Inform Design of a Preceptor Development Program in Jordan: A Multi Method Study

PONE-D-25-26793R2

Dear Dr. Albeitawi,

We’re pleased to inform you that your manuscript has been judged scientifically suitable for publication and will be formally accepted for publication once it meets all outstanding technical requirements.

Kind regards,

Christmal Dela Christmals, PhD, MSc, BSc, RN

Academic Editor

PLOS ONE

Additional Editor Comments (optional):

Congratulations on your excellent work.

Despite meeting all scientific requirements, it is crucial to proofread the paper. For example, the term "competency based" should be made a compound word (competency-based).
---

## [Editor Report · Acceptance letter]

PONE-D-25-26793R2

PLOS ONE

Dear Dr. Albeitawi,

I'm pleased to inform you that your manuscript has been deemed suitable for publication in PLOS ONE. Congratulations! Your manuscript is now being handed over to our production team.

Kind regards,

on behalf of

Professor Christmal Dela Christmals

Academic Editor

PLOS ONE